


1
2
3

# Study on the influence of seafloor soft soil layer on seismic ground motion

**Jingyan Lan[1], Juan Liu[2], Xing Song[1*]**

[1]Key Laboratory of Geomechanics and Geotechnical Engineering, Guilin University of Technology, Guilin, Guangxi 541004, China

[2]Institute of Engineering Mechanics, China Earthquake Administration, Harbin, Heilongjiang 150080, China

**Correspondence:** Xing Song (425270453@qq.com)

16

17

18

**Abstract.** In the complex medium system of sea area, the overlying sea water and the surface soft soil have a significant impact on the seafloor ground motion, which brings great seismic risk to the safety of offshore engineering structures. In this paper, four sets of typical free field models are constructed and established, which are land model, land model with surface soft soil, sea model and sea model with surface soft soil. The dynamic finite difference method is used to carry out two-dimensional seismic response analysis of typical free field based on the input forms about P and SV wave. By comparing the seismic response analysis results of four groups of calculation models, the effects of overlying seawater and soft soil on peak acceleration and acceleration response spectrum are studied. The results show that when SV wave is input, the peak acceleration and response spectrum of the surface of soft soil on the surface and the seabed surface can be amplified, while the overlying sea water can significantly reduce the ground motion. When P wave is used, the effect of overlying seawater and soft soil on peak acceleration and response spectrum of surface and seabed can be ignored. The peak acceleration decreases first and then increases from the bottom to the surface, and the difference of peak acceleration calculated by four free field models is not obvious. The results show that the overlying sea water and the surface soft soil layer have little effect on the peak acceleration of ground motion below the surface.



## 1 Introduction

Both empirical and theoretical studies of earthquake damage show that the site conditions, especially the soft soil site conditions, are the important factors that affect the aggravation of surface earthquake damage and the significant amplification of theoretical ground motion (Celebi, 1991; Huang et al., 2009; Kubo et al., 2019; Wang et al., 2019). With the development of offshore engineering in recent years, some of them, such as offshore oil platform, cross sea bridge, subsea tunnel, etc., have been developed rapidly. Therefore, dynamic characteristics of subsea soft soil and its influence on the ground motion are paid more and more attention. The current research mainly focuses on the simulation of land-based ground motion. In the seismic response analysis of the actual sea area engineering site, the influence of the self weight stress of the overlying sea water and the action of the overlying saturated soft soil are ignored (Fan et al., 2018), and the one-dimensional frequency domain or time domain seismic response analysis method consistent with the land area engineering is still used (Idriss and Sun, 1992; Streeter et al., 1974; Bardet et al., 2000; Hashash and Park, 2001; Bardet and Tobita, 2001), which is inconsistent with the actual sea area saturated two-dimensional complex medium system. Contemporary studies have shown that the pore water saturation of underwater soil layer has a great influence on the amplitude of vertical in-plane motion (Yang and Sato, 2000; Yang, 2001; Wang and Hao, 2002; Zhang et al., 2012). However, none of these studies considered the effect of the sea water layer on the seismic ground motion in the offshore environment. Boore and Smith (1999) analyzed the seismic records obtained from the undersea seismic survey system deployed off the coast. The theoretical calculation shows that the influence of the sea water layer on the horizontal component of the ground motion can be ignored. Since there are few records of the sea floor movement, some methods have been proposed to simulate the spatial changes of the sea floor movement (Meng, 2007; He et al., 2015). Petukhin et al. (2010) analyzed two real seabed models with and without sea water layer, and concluded that when the thickness of sea water is within 5 km, the effect of sea water layer on Rayleigh wave is significant; when the thickness of sea water is greater than 10 km, the effect of sea water on ground motion can be ignored. Nakamura et al. (2014) used the strong earthquake data recorded on the surface of the Kii Peninsula and near the Nankai Trough to study the abnormal large earthquake amplification in the seabed area. Zhang et al. (2019) deduced the analytical expression of the wave of the seabed foundation site when the plane P wave or SV wave incident, and thought that the water depth had little influence on the peak value of the horizontal displacement, but the resonance frequency increased with the increase of the thickness of the water layer.

In view of this, in order to study the influence of the self weight stress of the overlying sea water, based on the second development of the finite difference software FLAC3D, this paper constructs four typical two-dimensional dynamic calculation models of the free field, carries out two-dimensional seismic response analysis of the free field, systematically studies the influence of the self weight stress of the overlying sea water and the soft soil layer on the peak value and acceleration response spectrum of the ground motion, through comparative analysis The difference of seismic response analysis results of land free field in sea area. The research results are helpful to reveal the influence mechanism of the saturated soil site on the ground motion, and further deepen and enrich the research progress and achievements in this field.In


view of this, in order to study the influence of the self weight stress of the overlying sea water,
based on the second development of the finite difference software FLAC3D, this paper
constructs four typical two-dimensional dynamic calculation models of the free field, carries
out two-dimensional seismic response analysis of the free field, systematically studies the
influence of the self weight stress of the overlying sea water and the soft soil layer on the peak
value and acceleration response spectrum of the ground motion, through comparative analysis
The difference of seismic response analysis results of land free field in sea area. The research
results are helpful to reveal the influence mechanism of the saturated soil site on the ground
motion, and further deepen and enrich the research progress and achievements in this field.
**2    Brief introduction of seismic response method based on finite difference method**
The quadrilateral element is utilized to divide the dynamic calculation area, the Mohr
Coulomb criterion is used for the nonlinear constitutive model of soil, the free boundary is
used for the artificial boundary, and the Rayleigh damping is selected for the damping. The
dynamic equation (1), (2) are formed according to the node equilibrium condition. After the
earthquake load is input, the fast Lagrangian finite difference method is used to solve the
equation. The node velocity (3), (4) and node displacement (5), (6) are obtained, and then the
next node is calculated. The above process is reiterated until the end of calculation.

$$M_i \ddot{u}_i^t(t) + C_i \dot{u}_i^t(t) + K_i \ddot{u}_i^t(t) = F_{ix}(t) \tag{1}$$


$$M_i \ddot{v}_i^t(t) + C_i \dot{v}_i^t(t) + K_i \ddot{v}_i^t(t) = F_{iy}(t) \tag{2}$$

Where: $M$ is the mass matrix, $C$ is the damping matrix, $K$ is the stiffness matrix, $u_i$、
$v_i$ is the node displacement, $\dot{u}_i^t$、$\dot{v}_i^t$ is the node speed, $\ddot{u}_i^t$、$\ddot{v}_i^t$ is the node acceleration, $F_i(t)$
is the external force on the node.

$$\dot{u}_i^t(t + \frac{\Delta t}{2}) = \dot{u}_i^t(t - \frac{\Delta t}{2}) + \frac{\Delta t}{M} F_{ix} \tag{3}$$


$$\dot{v}_i^t(t + \frac{\Delta t}{2}) = \dot{v}_i^t(t - \frac{\Delta t}{2}) + \frac{\Delta t}{M} F_{iy} \tag{4}$$


$$u_i(t + \Delta t) = u_i(t) + \Delta t \dot{u}_i(t + \frac{\Delta t}{2}) \tag{5}$$


$$v_i(t + \Delta t) = v_i(t) + \Delta t \dot{v}_i(t + \frac{\Delta t}{2}) \tag{6}$$

**3    Dynamic calculation model**
Four groups of free field calculation models are designed and constructed, including land
layered site model, land layered site model with surface soft soil, sea layered site model and
sea layered site model with surface soft soil. The layer thickness and sea water depth of each
group of calculation models are presented in Figure 1. In order to effectively analyze the




influence of overlying sea water and soft soil layer on ground motion, the layered soil in the
four calculation models in this paper refers to the calculation model of Chen (2016), and in
order to simplify the calculation model and improve the simulation accuracy and efficiency,
this paper regards sea water as ideal fluid and ignores its viscosity, and realizes the influence
of overlying sea water self weight stress by setting dynamic water pressure, The horizontal
layered free field of seawater saturated seabed basement is established. Layered soil in the
four groups of calculation models uses the same physical and mechanical property parameters
(Lan et al., 2012), and the calculation results are shown in Table 1 by using formulas (7) to
(9).

$$G = V_S^2 \rho \tag{7}$$

$$V_P = \sqrt{\frac{\lambda + 2G + \alpha^2 M}{\rho}} = \sqrt{\frac{G(2 - 2\mu')/(1 - 2\mu') + \alpha^2 M}{\rho}} \tag{8}$$

$$K = V_P^2 \rho + \frac{4}{3} G \tag{9}$$

Where, $G$ is the shear modulus, $V_P$ is the compression wave velocity, $K$ is the bulk modulus,
$\mu'$ is the Poisson's ratio of soil skeleton, $\alpha$ is the compression parameter of soil, and $M$ is the
compression parameter of water.

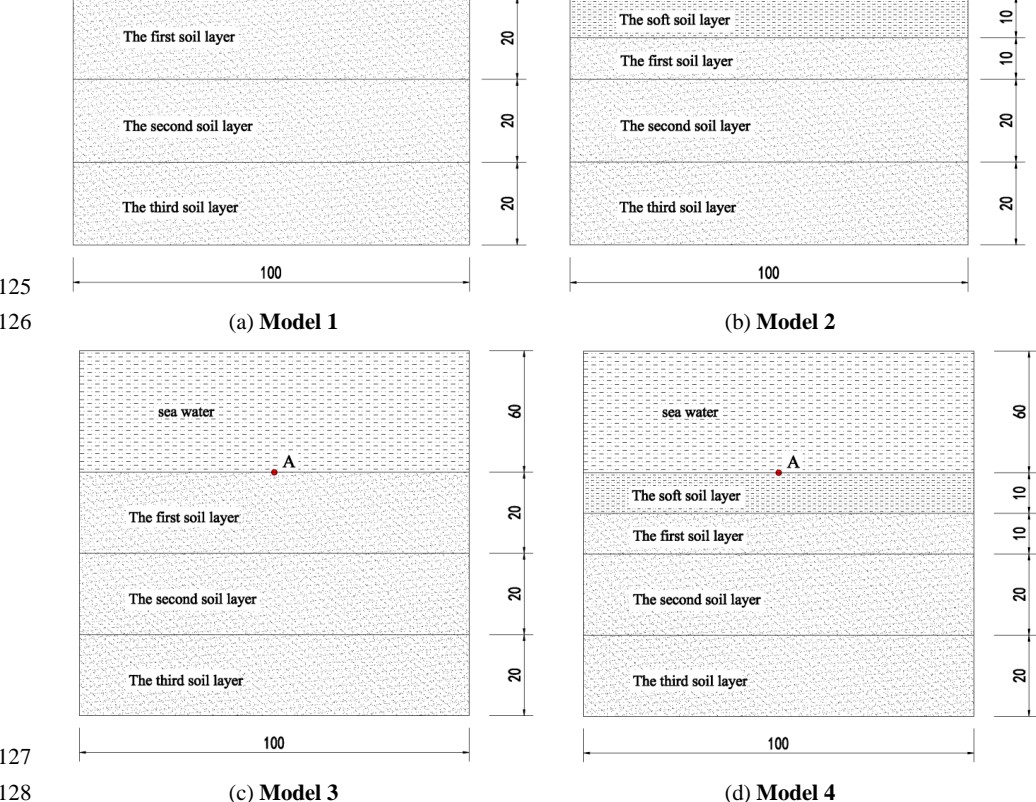

(a) **Model 1**

(b) **Model 2**

(c) **Model 3**

(d) **Model 4**

**Figure 1.** Four typical free field models
**Table 1.** Parameter of stratified layer model

| Basic properties | Soft soil layer | First soil layer | Second soil layer | Third soil layer |
|---|---|---|---|---|
| $\rho$(kg/m3) | 1500 | 1700 | 2000 | 2250 |
| $G$(MPa) | 48.6 | 97 | 221 | 951 |
| $K$(MPa) | 354 | 478 | 6130 | 9130 |
| $v_s$(m/s) | 180 | 240 | 360 | 650 |
| $v_p$(m/s) | 1550 | 1700 | 1800 | 2170 |
| $C$(kPa) | 10 | 10 | 10 | 6500 |
| $\varphi$(°) | 18 | 30 | 30 | 45 |

Four groups of free field calculation models (Figure 1) are meshed by FLAC3D. According
to the principle that the size of the grid element should not be greater than 1/10 of the
minimum wavelength in the input seismic wave, the wave propagation in the soil layer can be
more accurately simulated, i.e
$$\Delta l \leq (\tfrac{1}{10} \sim \tfrac{1}{8})\lambda \qquad (10)$$
$$\lambda = \frac{V}{f} \qquad (11)$$
Where, $\Delta l$ is the maximum grid size, $\lambda$ is the minimum input wavelength, $f$ is the highest
frequency of seismic wave, $V$ is the wave velocity of seismic wave.
By substituting the soil parameters into the above formula, we can get the value of $\Delta l$ =3m
in the form of SV wave input and $\Delta l$ =17m in the form of P wave input. In order to improve
the simulation accuracy, the mesh size of SV wave input is 2m×2m, and that of P wave input
is 5m×5m.
**4   Analysis of numerical simulation results of earthquake response**
**4.1 Base input ground motion**
Two representative natural ground motion time history, El Centro wave and Kobe wave, are
selected for the input seismic wave of basement. In order to enhance the calculation efficiency,
the time window of the input base time history is reduced, the first 20s of the time history
curve is intercepted and retained, and the amplitude is scaled at the same time. The two peak
values of seismic wave input are adjusted to 0.20g, which can simulate the seismic response
under the action of a medium strong earthquake. The acceleration time history curve of El
Centro wave and Kobe wave adjusted by the peak value and duration is given in Figure 2.

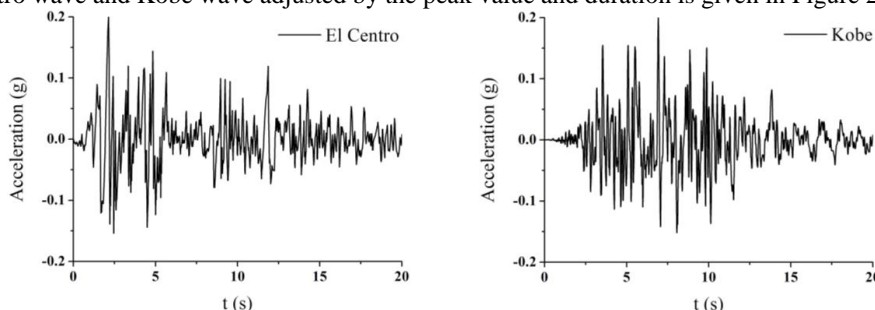

**Figure 2.**   Acceleration time history curve of seismic waves



## 4.2 Effect of overlying seawater and soft soil on peak acceleration and amplification coefficien

Taking El Centro wave and Kobe wave as the basic input, SV wave and P wave are used respectively, and the finite difference method introduced in Section 1 is used to carry out the numerical simulation of two-dimensional free field seismic response analysis. The peak acceleration of ground motion at a point A of the surface and seabed surface of four groups of free field models is calculated. At the same time, we define a peak amplification coefficient β, which is expressed as the ratio of the surface peak acceleration $PGA_{surface}$ to the base input peak acceleration of $PGA_{bedrock}$, i.e:

$$\beta = \frac{PGAsurface}{PGAbedrock} \qquad (12)$$

Then, we get the peak acceleration and peak amplification coefficient of point a, as showed in Table 2 and Table 3.

**Table 2.**  Peak acceleration of monitoring point A under different seismic wave input conditions

| Base input | Input form | Model 1 | Model 2 | Model 3 | Model 4 |
|---|---|---|---|---|---|
| El Centro | SV | 0.659 | 0.774 | 0.651 | 0.705 |
|  | P | 0.119 | 0.120 | 0.128 | 0.130 |
| Kobe | SV | 0.421 | 0.534 | 0.393 | 0.509 |
|  | P | 0.119 | 0.120 | 0.123 | 0.123 |

**Table.3**  Peak acceleration of monitoring point A under different seismic wave input conditions

| Base input | Input form | Model 1 | Model 2 | Model 3 | Model 4 |
|---|---|---|---|---|---|
| El Centro | SV | 3.29 | 3.87 | 3.26 | 3.57 |
|  | P | 0.59 | 0.60 | 0.64 | 0.65 |
| Kobe | SV | 2.10 | 2.67 | 1.97 | 2.55 |
|  | P | 0.59 | 0.60 | 0.61 | 0.62 |

It can be seen from the above results that when the seismic wave is input at the base in the form of SV wave, the peak acceleration and amplification coefficient β of point a of the four calculation models are obviously different. The results with the largest peak value and amplification coefficient are the land model with surface soft soil, then the sea model with surface soft soil, then the land model, and the sea model with the smallest result. Therefore, it can be concluded that the surface soft soil can amplify the peak acceleration of the surface and the seabed surface, while the overlying sea water can significantly reduce the ground motion. It can also be seen from the results in Table 2 and table 3 that under the same set of calculation model conditions, the calculation results for El Centro wave as the basic input is higher than those with Kobe wave input

When the seismic wave is input as a base in the form of P-wave, the peak acceleration and amplification coefficient β of point a of the four calculation models are almost the same. The main reason for this phenomenon may be that the overlying sea water and the self weight of the soil layer restrain the seismic response and amplification effect of P wave as vertical input. At the same time, because of the property of P wave and the direction of vibration propagation, the vertical seismic action of P wave is small. However, the SV wave is horizontal input and perpendicular to the propagation direction, which can cause soil shear



deformation. Furthermore, it can also be shown that in the seismic design and time history
analysis of practical projects, the shear effect of horizontal ground motion is still the primary
consideration and attention.
**4.3 Effect of overlying seawater and soft soil on acceleration response spectrum**
After calculation and analysis, the acceleration response spectra of four groups of free field
models at point A of the surface and seabed surface under the input of El Centro wave and
Kobe wave in the form of P and SV wave are given, as shown in Figure 3 and Figure 4.

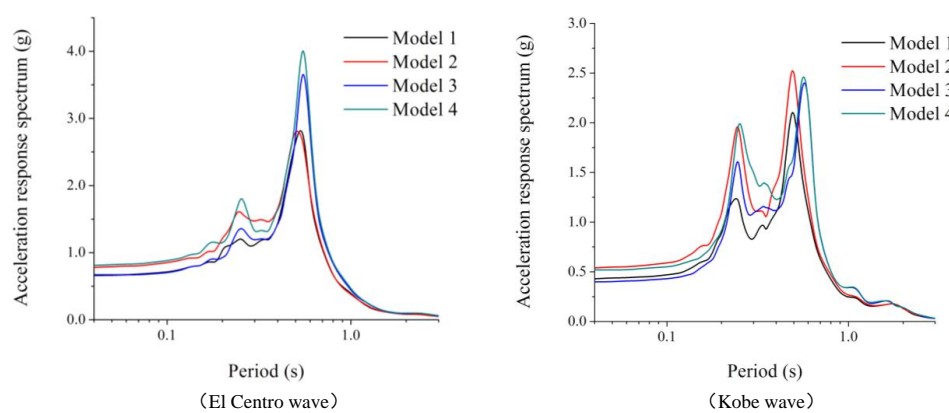

193                                (El Centro wave)                              (Kobe wave)
**Figure 3.**   Acceleration response spectrum of point a of four free field models in the form of SV input

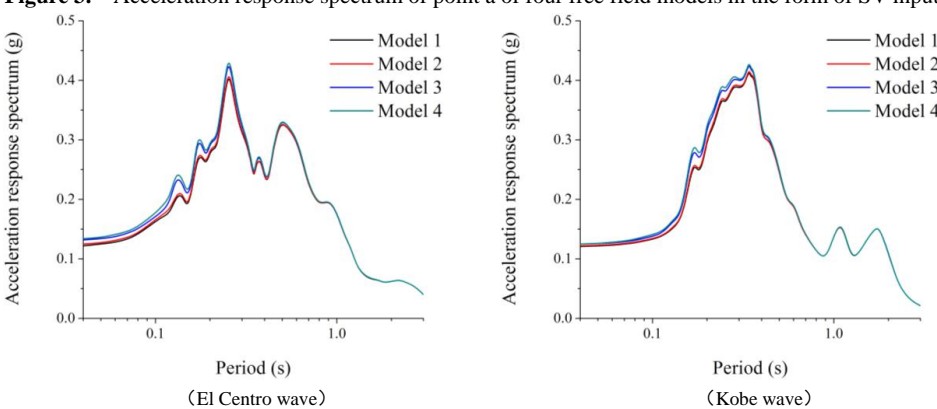

196                                (El Centro wave)                              (Kobe wave)
**Figure 4.**   Acceleration response spectrum of point a of four free field models in the form of P-wave input

It can be seen from Figure 3 that when El Centro wave is input in the form of SV wave,
the acceleration response spectrum reaches the maximum value when the period is 0.6s, and
the acceleration response spectrum results of the two groups of sea free field models are
generally higher than those of the land model, and the response spectrum results of model 4
are slightly higher than those of model 3. When the Kobe wave is input in the form of SV
wave, there are two obvious peaks in the acceleration response spectrum, and the long period
components of the acceleration response spectrum of the two groups of sea free field models


are generally higher than that of the land model.

**4.4 Variation trend of amplification effect of ground motion along depth**

In order to analyze the propagation characteristics and attenuation law of seismic wave from
the bottom of typical free field to the surface, in addition to monitoring point A of surface and
seabed surface, monitoring calculation points are set every 10m for the above four groups of
free field models, and the relationship curve of peak acceleration of four groups of free field
models with the depth of soil layer is given. Figure 5 shows the results of SV wave input, and
Figure 6 shows the results of P-wave input result. It can be seen from Figure 5 that the peak
acceleration of the ground motion from the bottom to the surface decreases first and then
increases, which shows that the ground motion changes at 50m underground and generally
enlarges at the surface, as shown in Table 4. At the same time, Figure 5 also shows that the
peak acceleration difference of four groups of free field models under the surface of 10 meters
is small. Because there is no difference in the calculation parameters and physical and
mechanical properties of the calculated model site soil, it shows that under the input condition
of SV wave, the impact of overlying sea water and surface soft soil layer on the peak
acceleration of ground motion under the surface is small.
For the case of P wave input, the difference of peak acceleration among the four groups of
free field models is small, and the peak acceleration decreases first and then increases from
the bottom to the surface, which shows that there is a change at 50m underground, and it
shows a reduction effect on the surface as a whole. Similarly, under the condition of P-wave
input, the effect of overlying sea water and soft soil layer on the peak acceleration of ground
motion under the surface is unimportant.

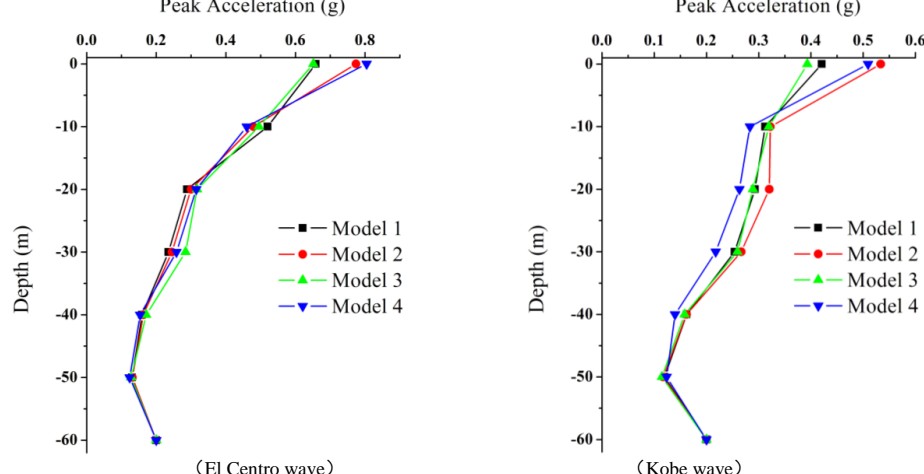

**Figure 5.**  Variation of peak acceleration with depth under SV wave input


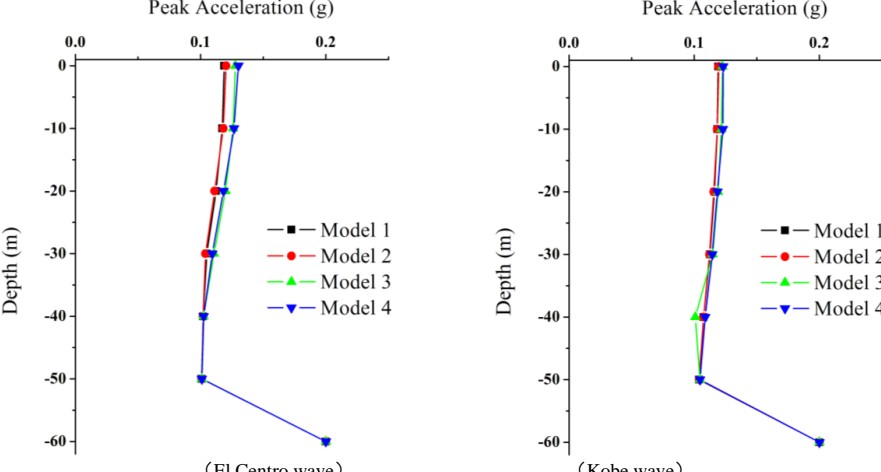

**Figure 6.**    Variation of peak acceleration with depth under P wave input
**5    conclusion**
In this paper, FLAC3D is used to construct four sets of typical free field calculation models.
Natural seismic waves of SV and P waves are used as the base input, and the finite difference
method is used to carry out two-dimensional seismic response analysis. Four sets of
calculation results of peak acceleration and acceleration response spectrum of typical free
field are obtained, and the following preliminary conclusions are summarized
(1) When the seismic wave is input in the form of SV wave, the surface soft soil can
amplify the peak acceleration of the surface and seabed surface, while the overlying sea water
can significantly reduce the ground motion. When the seismic wave is newly input as a base
in the form of P wave, the effect of overlying sea water and soft soil on the peak acceleration
of the surface and seabed surface is small and can be ignored.
(2) When El Centro wave is input as the base in the form of SV wave, the acceleration
response spectrum results of the two groups of sea free field models are generally higher than
those of the land model, and the response spectrum results of model IV are slightly higher
than those of model III. When the Kobe wave is input as the base in the form of SV wave, the
long period component of acceleration response spectrum of the two groups of sea free field
models is generally higher than that of the land model. Similar to the peak acceleration results,
when the seismic wave is input in the form of P wave, the difference between the four groups
of free field models is small.
(3) The peak acceleration decreases first and then increases from the bottom to the surface.
In addition to the results of the peak acceleration of the surface, the peak acceleration
difference between the layers of the underground obtained by each model is not obvious. For
the input of SV wave, the overlying seawater and soft soil layer have little effect on the peak
acceleration of ground motion below the surface, and show an amplification effect on the
whole at the surface. For the P-wave input, the difference of peak acceleration between the
underground layers is small, and the effect of reduction on the surface is overall.
In conclusion, through the comparative analysis of the two-dimensional seismic response



analysis results of four typical free field models, it shows that the overlying sea water and the
surface soft soil layer have certain influence on the ground motion, which is an important
factor that cannot be ignored in the actual engineering seismic design analysis and checking
calculation. Therefore, the conclusion of this paper is helpful to improve the understanding of
the dynamic characteristics of saturated soft soil and the amplification effect of site ground
motion, and has certain reference value for seismic design of sea area engineering.
*Data availability*. The data used to support the findings and results of this study are available
from the corresponding author upon request.
*Author contributions*. All authors (JL[1], JL[2], XS) contributed to this paper. JL[1] and JL[2]
discussed and wrote the original manuscript. JL[2] collected calculating data and established the
different models. JL[1] and XS ran the simulations, formal analysis, and data curation. JL[1] made
all the figures, and prepared the writing with contributions from all co-authors.
*Competing interests*. The authors declare that they have no conflict of interest.
*Financial support.* This research has been supported by the National Natural Science
Foundation of China (Grant No. 51408559), Guangxi Natural Science Foundation (Grant No.
2018GXNSFAA281183).

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
