# Peer review of "Study on the influence of seafloor soft soil layer on seismic ground"

_Natural Hazards and Earth System Sciences, 2020_

## Referee Comment (RC1) · Anonymous Referee #1 · 16 Sep 2020

The paper presents an interesting study related to the influence of seafloor soft soil and overlying sea water on ground motion. At present, most studies believe that water cannot propagate shear waves, and the influence of overlying sea water will not be considered in engineering seismic research. In this paper, it shows that the self-weight of overlying water has a significant effect on ground motion, so this paper has important scientific significance. However, it may be more meaningful to calculate and analyze the real undersea site instead of the artificial site models, which is of course a very difficult task. The article is well written, enjoyable to read and clear enough in the development of the subject. I found the manuscript interesting and worthy of publication in the NHESS journal.

I have just few minor further points. 1. In the line 67, the plane should be deleted. 2.

[Figure]

Figure 1 is not so clear, it would be better if the picture was changed to color. Especially the part of the sea water. 3. The spectral characteristics of the input acceleration time history should be supplemented in figure 2, such as Fourier spectrum, etc. 4. I am not very clear why the grid division in the calculation between P wave and SV wave is different, and there is no clear explanation in this paper.

---

## Referee Comment (RC2) · Anonymous Referee #2 · 31 Oct 2020

I have read with great interest the paper entitled: 'Study on the influence of seafloor soft layer on seismic ground motion', by Lan et al. The manuscript is very well written and organized. The authors summarized the previous research results carefully, and pointed out the existing problems of ground motion response analysis in sea area engineering. Several groups of representative sea area site models were calculated and analyzed by finite element method, and the effects of overlying sea water and soft soil surface on ground motion are analyzed in this paper. I think this study represents an important contribution to the seismic safety evaluation of marine engineering. I recommend publication with minor revisions. My minor suggestions are listed below: (1) What is the unit of length in Figure 1? I think it should be "meter", but it is not mentioned in that section. (2) Line110, the compression coefficient of overlying water and

[Figure]

Poisson's ratio of soil are mentioned in this section, which are important parameters for dynamic analysis calculation, but they are not given in this paper. (3) Line118, there are something wrong with the units in Table 1, for example, the units of stiffness and modulus should be kPa, rather than MPa. (4)Line156, the title of Table 3 is wrong, it should be the amplification factor. (5) It is suggested that the amplification factor ratio of response spectrum between the surface and the base input should be given, so that we can know in which period the amplification of the four models is the most significant. It is necessary to do proper analysis and discussion. (6) The Mohr-Coulomb constitutive model used in this paper is an ideal elastic-plastic model. It is suggested that the reason for using this constitutive relation should be discussed in this paper.

---

## Author Comment (AC1) · 20 Nov 2020

Dear Referee 1. We appreciate your positive comments on the manuscript. I will revise or explain the following three comments one by one. 1. The word "Plane" indicates the meaning of inputting seismic waves in a two-dimensional plane, and of course it is no problem to remove them. 2. We replot the figure 1, changed it to a color picture, and improved the quality. The new figure 1 is as follows: 3. The Fourier spectra of Kobe and El Centro waves are supplemented in this paper. The new figure 2 is as follows: 4. In this paper, the estimation formula of grid size is given in formula 10, $\Delta l \leq (1/10 \sim 1/8)\lambda$. The main reason for the difference between the grid input of SV wave and P wave is that the propagation velocity of SV wave and P wave is different, refer to formula 11, $\lambda = V/f$. Where, $\Delta l$ is the maximum grid size, $\lambda$ is the minimum

input wavelength, f is the highest frequency of seismic wave, V is the wave velocity of seismic wave. By substituting the soil parameters into the above formula, we can get the value of $\Delta l$=3m in the form of SV wave input and $\Delta l$=17m in the form of P wave input. In order to improve the simulation accuracy, the mesh size of SV wave input is 2m×2m, and that of P wave input is 5m×5m. Thank you for providing us with comments and suggestions on our manuscript.

Please also note the supplement to this comment:
https://nhess.copernicus.org/preprints/nhess-2020-177/nhess-2020-177-AC1-supplement.pdf
* * *
[Figure]

**Figure 1.** Four typical free field models

Fig. 1.

[Figure]

**Figure 2.** Acceleration time history curve of seismic waves

**Fig. 2.**

**Supplement:**

Dear Referee 1. We appreciate your positive comments on the manuscript. I will revise or explain the following three comments one by one.

1. The word "Plane" indicates the meaning of inputting seismic waves in a two-dimensional plane, and of course it is no problem to remove them.

2. We replot the figure 1, changed it to a color picture, and improved the quality. The new figure 1 is as follows:

[Figure]

**Figure 1.** Four typical free field models

3. The Fourier spectra of Kobe and El Centro waves are supplemented in this paper. The new figure 2 is as follows:

[Figure]

**Figure 2.** Acceleration time history curve of seismic waves

4. In this paper, the estimation formula of grid size is given in formula 10, $\Delta l \leq (\frac{1}{10} \sim \frac{1}{8})\lambda$. The main reason for the difference between the grid input of SV wave and P wave is that the propagation velocity of SV wave and P wave is different, refer to formula 11, $\lambda = \frac{V}{f}$. Where, $\Delta l$ is the maximum grid size, $\lambda$ is the minimum input wavelength, f is the highest frequency of seismic wave, V is the wave velocity of seismic wave. By substituting the soil parameters into the above formula, we can get the value of $\Delta l$ =3m in the form of SV wave input and $\Delta l$=17m in the form of P wave input. In order to improve the simulation accuracy, the mesh size of SV wave input is 2m×2m, and that of P wave input is 5m×5m.

Thank you for providing us with comments and suggestions on our manuscript.

---

## Author Comment (AC2) · 20 Nov 2020

Dear Referee 2. We appreciate your very useful and professional comments on the manuscript. I will revise or explain the following six comments one by one. 1. We left out the unit of figure 1 in the original manuscript, replot figure 1, and gave the unit "m" in the new manuscript. The new figure is as follows: 2. In this paper, it is assumed that water is an incompressible ideal fluid, so the compression coefficient of water is not needed in the calculation. The four calculation models in this paper are all saturated soil, so the Poisson's ratio of soil is 0.5, which has been supplemented in Table 1. 3. We double-checked the units of G and K, and it is true that their units are MPa, not kPa. 4. Yes, I'm sorry that we made a mistake about the title of Table 3, which has been revised now. 5. We agree with the reviewers and supplement the results of the

response spectral ratio between the surface and base acceleration, and the necessary analysis and discussion are given in this paper. The supplementary discussion is as follows: In order to investigate the amplification effect of the surface acceleration response spectrum in each period, the result of the acceleration response spectrum is divided by the base input response spectrum, and the result of the response spectral ratio is obtained, as shown in figures 5 and 6, in which the input form of ground motion in figure 5 is SV wave, and the input form of base ground motion in figure 6 is P wave. According to figure 3 and figure 5, it can be found that the maximum value of reaction spectrum ratio appears at the periodic point of the maximum value of response spectrum for the SV wave as input, and the amplification effect is very obvious, and the maximum amplification factor is about 8 to 10. As the result of the input form of P wave, we find that the period of the maximum acceleration response spectrum of figure 4 is inconsistent with that of the maximum response spectrum ratio of figure 6, and the result of the response spectrum ratio shows an undulating shape. Generally speaking, the surface response spectrum is not magnified compared with the base response spectrum, and shows a shrinking result in each period. 6. That is right, the Mohr-Coulomb constitutive model used in the manuscript is an ideal elastic-plastic model. That is why we choose the M-C model for constitutive model of soil layers. And another important reason is that the parameters of M-C model are relatively few and can be easily obtained in the laboratory tests, so it is widely used in the practical work of soil seismic response analysis. Thank you for providing us with comments and suggestions on our manuscript.

Please also note the supplement to this comment:
https://nhess.copernicus.org/preprints/nhess-2020-177/nhess-2020-177-AC2-supplement.pdf

The first soil layer
The second soil layer
The third soil layer
(a) Model 1
20m
20m
20m
100m

The soft soil layer
The first soil layer
The second soil layer
The third soil layer
(b) Model 2
10m 10m
20m
20m
100m

sea water
The first soil layer
The second soil layer
The third soil layer
(c) Model 3
60m
20m
20m
20m
100m

sea water
The soft soil layer
The first soil layer
The second soil layer
The third soil layer
(d) Model 4
60m
10m 10m
20m
20m
100m

**Figure 1.**   Four typical free field models

**Fig. 1.**

[Figure]

( El Centro wave )  ( Kobe wave )

**Figure 5.** Response spectral ratio between the surface and the base input of SV wave form

**Fig. 2.**

[Figure]

**Figure 6.** Response spectral ratio between the surface and the base input of P wave form

**Fig. 3.**